# Sequence-Specific Electrochemical Genosensor for Rapid Detection of *bla*_OXA-51-like_ Gene in *Acinetobacter baumannii*

**DOI:** 10.3390/microorganisms10071413

**Published:** 2022-07-13

**Authors:** Swarnaletchumi Kanapathy, Godwin Attah Obande, Candy Chuah, Rafidah Hanim Shueb, Chan Yean Yean, Kirnpal Kaur Banga Singh

**Affiliations:** 1Department of Medical Microbiology & Parasitology, School of Medical Sciences, Universiti Sains Malaysia, Health Campus, Kubang Kerian 16150, Kelantan, Malaysia; swarni_21@yahoo.com (S.K.); chuahcandy@uitm.edu.my (C.C.); hanimshueb@gmail.com (R.H.S.); yychan@usm.my (C.Y.Y.); 2Department of Microbiology, Faculty of Science, Federal University of Lafia, Lafia 950101, Nasarawa State, Nigeria; obandegodwins@gmail.com; 3Faculty of Health Sciences, Universiti Teknologi MARA, Kampus Bertam, Kepala Batas 13200, Penang, Malaysia

**Keywords:** genosensor, *Acinetobacter baumannii*, *bla*
_OXA-51-like_, antibiotic, nosocomial

## Abstract

*Acinetobacter baumannii* (*A. baumannii*) are phenotypically indistinguishable from the *Acinetobacter calcoaceticus*–*A. baumannii* (ACB) complex members using routine laboratory methods. Early diagnosis plays an important role in controlling *A. baumannii* infections and this could be assisted by the development of a rapid, yet sensitive diagnostic test. In this study, we developed an enzyme-based electrochemical genosensor for asymmetric PCR (aPCR) amplicon detection of the *bla*_OXA-51-like_ gene in *A. baumannii*. *A. baumannii*
*bla*_OXA-51-like_ gene PCR primers were designed, having the reverse primer modified at the 5′ end with FAM. A *bla*_OXA-51-like_ gene sequence-specific biotin labelled capture probe was designed and immobilized using a synthetic oligomer (FAM-labelled) deposited on the working electrode of a streptavidin-modified, screen-printed carbon electrode (SPCE). The *zot* gene was used as an internal control with biotin and FAM labelled as forward and reverse primers, respectively. The *bla*_OXA-51-like_ gene was amplified using asymmetric PCR (aPCR) to generate single-stranded amplicons that were detected using the designed SPCE. The amperometric current response was detected with a peroxidase-conjugated, anti-fluorescein antibody. The assay was tested using reference and clinical *A. baumannii* strains and other nosocomial bacteria. The analytical sensitivity of the assay at the genomic level and bacterial cell level was 0.5 pg/mL (1.443 µA) and 10^3^ CFU/mL, respectively. The assay was 100% specific and sensitive for *A. baumannii*. Based on accelerated stability performance, the developed genosensor was stable for 1.6 years when stored at 4 °C and up to 28 days at >25 °C. The developed electrochemical genosensor is specific and sensitive and could be useful for rapid, accurate diagnosis of *A. baumannii* infections even in temperate regions.

## 1. Introduction

*Acinetobacter baumannii* has emerged as an important human health risk that is associated with nosocomial infections such as pneumonia, bacteraemia, urinary tract infection, and meningitis [1,2,3]. Infections due to *A. baumannii* are common among immunocompromised patients who are critically ill in intensive care units (ICUs) or among patients who have undergone major surgical procedures [4,5]. A major challenge faced by many clinicians is treatment options for patients infected with extensively multi-drug-resistant *A. baumannii*. Currently, there are limited antibiotic options available for the treatment of these infections, hence making *A. baumannii* infections a grave concern for public health worldwide [6,7,8].

Within a few decades, *A. baumannii* has demonstrated a remarkable ability to rapidly develop resistance against multiple antibiotics [1,9]. Using various mechanisms, multi-drug-resistant *A. baumannii* (MDRAB) exhibit resistance to several existing antibiotics, including β-lactams, fluoroquinolones, tetracyclines, and aminoglycosides [10,11,12]. Carbapenems are among the drugs of choice for treating nosocomial infections [13] but their efficiency has been increasingly compromised by the spread of carbapenem-resistant isolates, mostly following acquisition of Class D carbapenemases. Studies have shown that *A. baumannii* isolates with carbapenem resistance tend to be resistant to all classes of antimicrobials except polymyxins and tigecycline in some cases [8,14]. Though previously effective against MDRAB, resistance to polymyxins have been reported in some clinical strains classified as extensively resistant (XDR) or pan-resistant *A. baumannii* [15,16]. The most common mechanism of carbapenem resistance in *A. baumannii* is through the production of carbapenem-hydrolysing beta-lactamases [17]. Class D β-lactamases are the most prevalent carbapenemases in *A. baumannii* isolates [18,19]. In addition to acquired carbapenem-hydrolysing class D oxacillinase (CHDL) gene clusters, which are present either in the chromosome or in the plasmids of *A. baumannii* strains, and known mainly as *bla*_OXA-23_-, *bla*_OXA-24/40_-, and *bla*_OXA-58-like_ genes [20], the chromosomal *bla*_OXA-51-like_ gene, which is intrinsic to *A. baumannii* isolates, confers carbapenem resistance when an ISAba1 element is inserted upstream of the gene [21,22].

Antibiotic resistant pathogens contribute to increased cost of healthcare and prolonged hospitalizations [23]. The use of ineffective antibiotics results in over usage, which further leads to an increase in the already mounting challenge of drug resistance among pathogens, thereby making them more difficult to treat [24]. Hence, in addition to accurate diagnosis of *A. baumannii* infections, rapid differentiation of *A. baumannii* from other *Acinetobacter* species in clinical settings would significantly improve patient outcomes. *A. baumannii* infections are typically diagnosed in routine laboratory using conventional culture method and biochemical tests. However, these methods are time-consuming and labour-intensive, rendering them less effective for rapid diagnosis. Due to its major health threat, there is a growing demand for a rapid detection test for *A. baumannii*. Among emerging technologies available, DNA biosensor, an analytical device incorporating a single-stranded oligonucleotide (probe) linked with a physicochemical transducer, offers an interesting alternative test. In recent years, this technology has been studied widely as a potential novel method for the detection of DNA hybridization in various fields such as the diagnosis of diseases including cancer [25,26,27], the detection of infectious agents [28,29], drug screening [30,31,32], crops screening [33,34], and forensic applications [35,36].

Electrochemical biosensors are specific, sensitive, and rapid, making them suitable for identification of microorganisms in samples [37]. DNA biosensors (genosensors) are inherently stable physicochemically and rely on the distinctive nature of genetic information to specifically identify microorganisms in human infections [38]. Genosensors employ immobilized DNA (or RNA) probes on a physical transducer to detect a target with a complementary sequence to the probe, through hybridization. Biological signals generated by this interaction are then detected as electrical signals through transducers, using appropriate equipment. Target analytes can be detected using biosensors via indirect sensing (labelled system) or direct sensing (label-free system) [37,39]. An electrodeposited gold nanostructure-based electrochemical biosensor was reported for *Enterococcus faecalis*, which could detect 30.1 ng µL^−1^ genomic DNA [40]. A few works have also reported biosensors for *A. baumannii* detection. Yeh and co-workers [41] developed an electro-microchip system based on DNA hybridization of a PCR-amplified *A. baumannii* target using biotin-labelled primers and gold-streptavidin nanoparticles. An electrochemical biosensor that used a gold electrode labelled with electroactive β-cyclodextrin (β-CD) for *A. baumannii* detection has also been reported. The electrochemical signals generated by the reduction of β-CD were recorded using differential pulse voltammetry (DPV) [42]. A more recent effort by Roushani and co-workers [43] showed the detection of *A. baumannii* from human serum samples using a molecularly imprinted polymer (MIP) on a glassy carbon electrode with electropolymerization of a dopamine monomer and *A. baumannii* template.

The use of screen-printed electrodes in biosensors confer the advantage of adaptability, ease of mass production, and selective specificity for target analytes, making biosensors suitable for rapid analysis on-site [44]. Additionally, among available transducer types, electrochemical biosensors have the advantage of having a high sensitivity, low detection limits, and flexibility for easy miniaturization [45]. Although some progress has been made towards accurate detection methods for detecting an important pathogen like *A. baumannii*, this study aimed to present an alternative detection for *A. baumannii* that is rapid, adaptable, and cost-effective, with applicability in both temperate and nontemperate climates. In this study, we developed a sequence-specific enzyme-based electrochemical genosensor assay for the detection of *A. baumannii* using a *bla*_OXA-51-like_ gene as the target gene and a screen-printed carbon electrode. The adaptability and cost-effective nature of carbon electrodes in biosensor development is advantageous to this method. To facilitate more accurate detection, the *zot* gene was used as an internal control. We further evaluated the analytical sensitivity, specificity, and diagnostic performance of this enzyme-based electrochemical DNA biosensor. In addition, to determine the shelf-life of the assay at various temperatures, an accelerated stability evaluation of the enzyme-based, electrochemical, sequence-specific biosensor assay was performed. This novel rapid test may facilitate the early detection of infections caused by *A. baumannii* and consequently help doctors to make prompt decisions about appropriate antibiotic treatment for infected patients.

## 2. Materials and Methods

### 2.1. Bacterial Strains

Reference strains of *A. baumannii* ATCC 19606 and *Shigella sonnei* ATCC 25931, which served as positive and negative control templates, respectively, were used for optimization of aPCR protocol and the development of an enzyme-based electrochemical DNA biosensor. A total of 76 clinical isolates were used in this study comprising 42 *A. baumannii* strains and 34 non-*Acinetobacter* bacterial pathogens (Table 1 and Table 2).

### 2.2. Bacterial Culture and Growth

All isolates used in this study were maintained as stock culture in 15% glycerol and stored at −70 °C. For the working culture, the bacterial strains were revived from glycerol stock culture by inoculating a loop of culture into tryptone soy broth (TSB) and incubating it overnight at 37 °C. Each overnight culture was then sub-cultured overnight onto blood agar (for purity confirmation), MacConkey agar (for confirmation of the strain), and nutrient agar (NA; for lysate preparation) and incubated overnight at 37 °C.

### 2.3. Development and Pretreatment of Screen-Printed Carbon Electrode (SPCE)

A screen-printed carbon electrode (SPCEs) was designed, and its fabrication was outsourced to a local company. The protocol was adapted from a previous study [46] with some modifications. The surface of SPCE was pre-washed with 100 µL of deionised water for 2 min (min). Subsequently, 5 µL of covalent agent [200 mM 1-ethyl-3-(3-dimethylaminopropyl)-carbodiimide (EDAC), and 50 mM N-hydroxysuccinimide (NHS)] were applied on the working electrode (WE) for 10 min to activate the surface. This process allowed for hybridization between the biotinylated capture probe and the synthetic oligomer on the surface of the working electrode. Five microliters (µL) of 0.05 mg/mL streptavidin were then coated onto the WE. Next, 50 µL of 1 M ethanolamine hydrochloride was applied and incubated in the dark for 10 min to inactivate the surface of SPCEs. The carbon surface was then blocked with 50 µL of 3% bovine serum albumin (BSA). SPCE was washed with deionized water by dipping once after each incubation step. The development of the SPCE until the BSA application step was used for internal control (IC) gene detection, whereas SPCEs for target gene detection were immobilized with 5 µL of biotinylated capture probe after the washing of BSA. SPCEs were then incubated for 10 min at room temperature, followed by washing with phosphate buffer saline (PBS).

### 2.4. Preparation of DNA Samples for PCR Amplification

#### 2.4.1. Lysate DNA Preparation

Lysate DNA was prepared by a boiling method. A single colony from an overnight culture on an NA plate was inoculated into 30 µL of water and boiled for 10 min. The lysate mixture was then centrifuged at 6000× *g* for 3 min and the supernatant containing the DNA was used for DNA templates during PCR amplification.

#### 2.4.2. Genomic DNA Preparation

A single bacterial colony from an overnight culture was inoculated into 10 mL of TSB and incubated at 37 °C overnight. Cells were harvested on the following day by centrifuging at 8000× *g* for 5 min; the supernatant was discarded. A cell pellet was collected and bacterial genomic DNA was extracted using a QIAamp DNA Mini Kit^®^ (Qiagen, Hilden, Germany), following the manufacturer’s instructions. The concentration of the purified genomic DNA was determined using a UV-VIS Biophotometer (Eppendorf, Hamburg, Germany) and stored at −20 °C.

#### 2.4.3. Preparation of Internal Control (IC) Plasmid DNA

TOP10 *E. coli* cells were treated with chemicals (magnesium chloride and calcium chloride) to become competent cells. These chemically competent TOP10 cells were either used directly for transformation or preserved in 15% glycerol stock. IC plasmid was obtained from previously prepared laboratory stock. The transformation of plasmid into competent cells was performed using the heat-shock method. The screening of clones with desired DNA inserts was carried out, and the presence of the IC gene (*zot*) was checked by performing a standard PCR. IC plasmid was extracted from bacterial clones using NucleoSpin^®®^ PlasmidQuickPure Kit (Macherey-Nagel, Düren, Germany), following the manufacturer’s instructions. The concentration of the purified IC recombinant plasmid was quantified using a UV-VIS Biophotometer (Eppendorf, Germany), and stored at −20 °C as a concentrated stock.

### 2.5. Preparation of aPCR Reaction Mixture

A pair of specific primers (Table 3) was designed to amplify a 135 bp-DNA sequence from the *A. baumannii bla*_OXA-51-like_ gene. The employed primer pair was in an optimal 1:20 forward-to-reverse-primer concentration ratio for aPCR amplification of the target ssDNA. Twenty microliter aliquots of the aPCR mix containing 1 × Taq buffer, 2.0 mM of MgCl_2_, 200 µM of dNTP mix, 1.0 U of *Taq* DNA polymerase, 0.1 µM of *bla*_OXA-51__F forward primer, and 1.0 µM of *bla*_OXA-51__R reverse primer were prepared in 0.2 mL PCR tubes using PCR-grade water. The reaction mix was then amplified using the following parameters: 3 min at 95 °C followed by 35 cycles of 95 °C for 30 seconds (s), 61 °C for 30 s, and 72 °C for 30 s. The amplification was further incubated for another 30 s at 61 °C and 5 min at 72 °C to extend any incomplete amplicons. The obtained aPCR amplicons were used directly for amperometric detection without any pretreatment or purification steps.

### 2.6. Electrochemical Detection of Synthetic DNA and Amplicons

A complementary biotinylated capture probe (Table 3) was used for *bla_OXA-51-like_* gene detection. The biotin-labelled capture probe was immobilized on the WE electrode surface with one end free to capture the synthetic *bla*_OXA-51-like_ target. After the washing step, 1 µM of the synthetic target was diluted with an equal volume of 4× sodium saline citrate (SSC) buffer and was applied on the WE surface. The SPCEs were then incubated for 20 min at room temperature. This process would allow hybridization between the biotinylated capture probe and the synthetic oligomer on the surface of the WE. Following this, the electrode was washed again with PBS to remove the unbound synthetic target. Five microliters of anti-fluorescein antibody in the ratio of 1:200 was applied onto the electrode surface and incubated for 5 min to allow binding of the enzyme with the fluorescein-labelled synthetic oligomer. The electrode was then washed with PBS prior to the addition of 70 µL of TMB:H_2_O_2_ (1:9) substrate onto the sensing area of SPCE. The electrode was subjected to 5 s incubation at 0 V standby potential, followed by amperometric measurement at −0.2 V for 60 secs with an interval time of 0.2 s. The current value at the end of the measurement period was recorded. For this electrochemical genosensor assay, aPCR technique was performed to generate single-strand DNA of the target *bla*_OXA-51-like_ gene for hybridization with the complementary capture probe, whereas the internal control (IC) gene was produced in double-stranded form for streptavidin–biotin binding. Figure 1 shows the schematic diagram for single-strand target *bla*_OXA-51-like_ gene detection by the electrochemical genosensor assay.

### 2.7. Analytical Evaluation of the Electrochemical Genosensor

#### 2.7.1. Analytical Specificity

The analytical specificity of the electrochemical genosensor assay was conducted using 30 bacterial strains made up of 6 *A. baumannii* reference strains, 8 *A. baumannii* clinical strains, 5 other *Acinetobacter* species, and 11 other pathogens including nosocomial pathogens (Table 1 and Table 2). DNA obtained by lysate extraction was used as a template for PCR amplification. The analytical specificity of the electrochemical genosensor assay was compared to the standard PCR amplification used as the gold standard. The electrochemical detection was conducted in triplicate (*n* = 3) for each DNA sample. The current signal obtained from this experiment was used to determine the cut-off value for the genosensor assay. The cut-off value for the positive results was calculated to be greater than or equal to the mean of the current plus three times the standard deviation (BG + 3SD) of the current signals for the negative control and non-target samples [47].

#### 2.7.2. Analytical Sensitivity at DNA and Cell Levels

The analytical sensitivity is the lowest concentration of purified genomic DNA required to give greater or equal current value to the cut-off point as determined in the analytical specificity. Purified genomic DNA of *A. baumannii* (ATCC 19606) was used for the genomic DNA level sensitivity evaluation. Serial dilution ranging from 0.05 pg to 1000 pg of genomic DNA was made using PCR-grade water. For the evaluation, 1 µL of the gDNA was used in the presence of 5 pg/µL of plasmid DNA in each reaction. The analytical sensitivity of the electrochemical genosensor was then compared to that from standard PCR amplification. The electrochemical detection was conducted in triplicates. 

For the determination of sensitivity at the bacterial cell level, bacterial stock culture was prepared by inoculating a single colony of the *A. baumannii* (ATCC 19606) overnight culture into TSB broth and incubating it at 37 °C overnight. After at least 18 h, 10-fold serial dilutions were made in 0.9% NaCl and 1 mL of each dilution was washed twice with deionized water by centrifugation at 8000× *g* for 10 min. The pellet was then re-suspended in 100 µL water. The cell suspension was boiled for 10 min and 2 µL of cell lysate were used as a template for aPCR. The colony count for each dilution was checked in parallel by plating on TSA. 

### 2.8. Clinical Application of the Electrochemical Genosensor

#### 2.8.1. Calculation of Sample Size for Spiked Blood Samples

In this study, blood samples were spiked with reference strains and clinical isolates as listed in Table 1 and Table 2. Evaluation with clinical blood samples was not carried out due to limited availability of samples during the study period. The sample size for spiked blood samples was calculated according to a previously published study. Based on the calculation, at least 35 spiked samples were required for each positive and negative sample evaluation. In this study, 48 positive strains consisting of 6 reference strains and 42 clinical isolates of *A. baumannii,* and 40 negative clinical isolates consisting of nosocomial and other pathogens, were spiked in blood samples and subsequently used for the diagnostic evaluation (Table 1 and Table 2).

#### 2.8.2. Preparation of Spiked Blood Sample

Spiked blood sample suspensions were prepared according to a previously described method with slight modifications. An overnight bacterial culture in TSB was diluted with 0.9% NaCl. Approximately 1 mL of 10^4^ CFU/mL of bacteria was spiked in 1 mL of blood culture. The pellet was then washed twice with 0.9% NaCl and re-suspended in 100 mmol/L Tris-HCl buffer (pH 8.0). The suspension was then boiled for 10 min, and the resulting supernatant was used as a template for aPCR and the products were subsequently detected using an electrochemical genosensor assay. To determine the diagnostic performance of spiked blood samples, the obtained results were further analysed to determine the diagnostic sensitivity, specificity, positive predictive value (PPV), and negative predictive value (NPV).

### 2.9. Stability Evaluation of the Modified SPCEs

The method used to evaluate the stability of the modified SPCE was adapted from a previous study [46] with slight modification. The stability evaluation test was carried out for 28 days at 1-week intervals. The SPCEs for IC detection were modified with streptavidin, whereas for the target gene, the SPCEs were modified with a capture probe before being stored at three different temperatures (4 °C, 25 °C, and 37 °C). Trehalose at 3% and 6% concentration was used as a stabilizer and was added onto the modified SPCEs’ surfaces to retain the streptavidin and capture probe activity on the electrode surface. The prepared trehalose was applied onto the surface of SPCEs. The electrodes were then freeze-dried for 15 min using a Heto vacuum concentrator (Thermo Scientific Heto, Denmark) connected to a LyoLab 3000 freeze-dryer (Thermo Scientific Heto, Denmark). The SPCEs were then placed in an aluminium pouch containing silica gel desiccants and stored at three different temperatures (4 °C, 25 °C, and 37 °C). The stability of the stored SPCEs was evaluated on days 1, 7, 14, 21, and 28.

## 3. Results

### 3.1. Determination of Amperometric Cut-Off Value

The cut-off value was calculated based on the mean of the amperometric current plus three times the standard deviation of BG (background control without using bacterial strains) and non-target samples. Based on this analytical evaluation, the cut-off value for this genosensor was 0.618 µA [0.369 µA + 3(0.083)]. Tests with ≥0.618 µA were interpreted as positive, whereas those below this current value were regarded as negative. The negative results obtained, however, were validated with a positive current response for IC. This was to ensure that inhibition did not occur during aPCR amplification, and hence, all negative results were indeed true negative results.

### 3.2. Evaluation of Analytical Specificity of Enzyme-Based Electrochemical DNA Biosensor

In this study, the interpretation of results was facilitated through the determination of a cut-off value for *bla*_OXA-51-like_ gene detection. As shown in Figure 2, the developed genosensor was able to distinguish positive samples from negative samples. The developed assay gave high amperometric signals to all *A. baumannii* tested, while a low background current was obtained with non-*A. baumannii* bacteria (negative samples). The negative results were validated with the presence of a high amperometric signal for IC.

### 3.3. Evaluation of Analytical Sensitivity of Enzyme-Based Electrochemical DNA Biosensor

The developed enzyme-based electrochemical genosensor assay was evaluated for its sensitivity at a genomic and bacterial level. This assay was also compared to the conventional PCR amplification with a 1:1 primer ratio through agarose gel electrophoresis analysis.

#### 3.3.1. Limit of Detection (LoD) at Genomic DNA Level

The analytical sensitivity (lowest limit of detection) of the enzyme-based electrochemical genosensor assay was evaluated and compared with conventional PCR using purified genomic DNA from *A. baumannii* (ATCC 19606) at concentrations varying from 0.05 pg to 1000 pg of genomic DNA. The LoD for conventional PCR using agarose gel electrophoresis analysis was 5 pg of genomic DNA (Figure 3a). As shown in Figure 3b, the amperometric current response of the target gene was directly proportional to the amount of genomic DNA. However, a plateau of the current response was observed at 100 pg of genomic DNA onwards. Interestingly, the LoD for this enzyme-based electrochemical genosensor assay at the genomic DNA level was 0.5 pg, with an amperometric current of 1.443 µA. Based on the result, this enzyme-based electrochemical genosensor assay was ten times more sensitive than the conventional PCR.

#### 3.3.2. Limit of Detection (LoD) at Bacterial Cell Level

The sensitivity analysis of the developed enzyme-based electrochemical genosensor assay was also performed at the bacterial cell level. Serially diluted bacterial lysates extracted from a pure culture of *A. baumannii* (ATCC 19606) were used to perform the sensitivity analysis. The bacterial concentrations ranging from 10^1^ to 10^7^ CFU/mL were tested. The sensitivity was tested at the bacterial cell level with the developed enzyme-based DNA biosensor and agarose gel analysis; both are shown in Figure 4. The detection limit at the bacterial level was found to be 10^3^ CFU/mL based on the calculated cut-off point. In contrast, the LoD for conventional PCR amplification (1:1 primer ratio) was 10^4^ CFU/mL. 

### 3.4. Diagnostic Evaluation of Enzyme-Based Electrochemical DNA Biosensor Assay Using Spiked Blood Samples

Diagnostic evaluation was carried out to test the potential use of the developed enzyme-based electrochemical genosensor for sequence-specific detection of the targeted microorganisms in clinical samples. Clinical evaluation was performed using blood samples spiked with a total of 88 bacterial strains which consisted of 48 positive samples (*A. baumannii* strains) and 40 negative samples (nosocomial and other pathogens). The results were interpreted based on the previously established cut-off value (0.618 µA). As shown in Figure 5, all positive *A. baumannii* clinical samples gave current signals above the cut-off value of 0.618 µA. On the other hand, all 40 negative samples were correctly interpreted as negative given that the current signals produced by the enzyme-based genosensor assay were all below 0.618 µA. All these negative results, however, had a positive signal for IC, indicating that these were true negative results.

Diagnostic evaluation results were compared to conventional PCR as the gold standard method. All the *A. baumannii* bacterial strains used in this study were accurately identified as true *A. baumannii* by amplified 16S rRNA gene restriction analysis (ARDRA), which was performed prior to this study. Clinical evaluation using spiked blood samples showed that the developed enzyme-based electrochemical genosensor correctly identified the *bla*_OXA-51-like_ gene in *A. baumannii* strains (48 positive and 40 negative samples) without cross-reaction with other pathogens.

### 3.5. Accelerated Stability Evaluation of Enzyme-Based Electrochemical DNA Biosensor

As shown in Figure 6, the SPCEs were stable for 28 days when stored at all the tested temperatures. SPCEs for the target gene detection (modified with the capture probe) were more stable than the SPCEs modified with streptavidin for IC detection. The signal produced by the target gene was demonstrated to be approximately similar throughout the 28 days. The results showed that SPCEs were stable at both 25 °C and 37 °C, indicating that the modified SPCEs can be kept at room temperature. The optimal concentration of stabilizer needed to preserve the modified SPCEs surface was shown to be 6% trehalose, as the amperometric signals for the target gene were more stable compared to the amperometric signal obtained using 3% trehalose.

According to Zheng et al. [20], one-day test kit storage at 37 °C is estimated to be equivalent to 21 days’ storage at 4 °C. Therefore, the stability of the modified SPCE at 37 °C ± 2 °C was calculated as the following:

= 28 days × 21

= 588 days at 4 °C

Hence, this result indicated that the modified SPCEs have an estimated minimum of 1.6 years of shelf-life when stored at 4 °C.

## 4. Discussion

The emergence of antibiotic-resistant bacteria causing infection in humans is a major global concern [48]. *A. baumannii* is one of the important pathogens that is included in this antibiotic-resistant bacteria group and is responsible for many hospital-acquired infections [6,9]. Infections due to *A. baumannii* are commonly reported in ICUs, particularly in immunocompromised patients who are critically ill [49,50]. *A. baumannii* is well-known for its resistance to multiple classes of antibiotics [51]. Carbapenem resistance, particularly class D carbapenemase genes, are of concern, as these genes are observed more frequently in *A. baumannii* strains. *A. baumannii* possesses a wide range of resistance mechanisms towards carbapenem such as causing changes in penicillin-binding proteins and alterations in the structure and the activity of efflux pumps [2,17]. Hence, *A. baumannii* infections have been reported to contribute to high mortality among patients [6,49,52].

In routine diagnostics, the identification of *A. baumannii* is performed using a panel of biochemical tests or a commercially available identification system such as the API 20NE system and VITEK. This conventional identification method is time-consuming and laborious. The detection of nucleic acid from microbial pathogens has also become an alternative detection method for the identification of causative agents. The development of PCR-based assays for the detection of *A. baumannii* has been described in several studies [53,54,55]. A real-time PCR assay has also been developed for *A. baumannii* detection [56], which provides faster results compared with conventional PCR. However, PCR assays require the use of expensive equipment and thus may not be an ideal solution for some countries [57,58]. Accurate and early detection of *A. baumannii* infections is very essential to efficient healthcare for patients due to the high mortality rate associated with its infections [59]. In recent years, various types of electrochemical biosensors based on the detection of bacterial nucleic acid have also been developed to replace PCR and gel electrophoresis analysis. DNA biosensors have been developed against several infectious diseases such as tuberculosis [60], hepatitis [61], dengue [62], and other food-borne diseases like *Escherichia coli* and *Salmonella typhimurium* [63]. DNA hybridization is a widely used technique for biosensor assay [64]. This process involves hybridization between single-stranded oligonucleotides and a complementary target sequence. The present study describes the development of an enzyme-based electrochemical genosensor for sequence-specific detection of the *bla*_OXA-51-like_ gene in *A. baumannii*. This genosensor assay was used as a detection method for aPCR amplicons instead of gel electrophoresis analysis. The hybridization process occurred after the immobilization of the single-stranded capture probe on the electrode surface, where the probe identified the complementary target gene amplified by aPCR, and formed a DNA probe-target hybrid. This hybridization mechanism is known as the direct hybridization method for target DNA detection in biosensor assay. This sequence-specific detection approach has been widely used due to its effectiveness and specificity even in the presence of non-complementary sequences [65,66]. Readable electrochemical signals are then generated through the oxidation of TMB with H_2_O_2_ reduction through a reaction catalyzed by an HRP enzyme [67,68]. Some DNA biosensors have been previously developed for *A. baumannii* detection. The work of Yeh and co-workers [41] also applied the hybridization principle, but incorporated an electro-microchip into their design and gold-streptavidin nanoparticles plus Ag+ -hydroquinone solution to enhance detection. The LoD observed at the genomic and bacterial level showed that the developed genosensor is 10 times more sensitive than a conventional PCR assay with an LoD of 5 pg. The LoD at the genomic level for our genosensor is slightly lower than the 0.825 ng mL^−1^ (1.2 fM) reported by Yeh and colleagues [41]. LoDs of 0.14 nM [42] and 1.86 nM [45] have also been provided with previously reported *A. baumannii* biosensors. Previous studies have also reported a similar LoD for aPCR amplicons of *E. coli* O157:H7 using gold nanoparticles [69] and *V. cholerae* using a magnetogenosensing assay [47].

While SPCE was used in this study for the development of an electrochemical genosensor, Eksin and co-workers [45] instead used chitosan as the material for the electrode. Although chitosan has been reported as a good natural polysaccharide for biosensor preparation [70,71,72], the use of screen-printed electrodes (SPEs) in biosensors has also been reported to confer the advantages of adaptability, low-cost, ease of mass production, and selective specificity for target analytes [44,73,74,75]. Additionally, SPEs can be easily customized in terms of their shape, substrate, and dimension, which provides for easy selectivity and calibration of SPE-based biosensors. Valuable analytical properties of biosensors such as specificity, sensitivity, accuracy, and reproducibility again can be easily achieved when SPEs surfaces are modified with nanomaterials [76,77].

Previous studies using PCR and other detection methods have shown that all *A. baumannii* isolates possessed a chromosomally located *bla*_OXA-51-like_ gene. Therefore, the detection of the *bla*_OXA-51-like_ gene can be used as a feasible and reliable way to identify *A. baumannii* from clinical and other specimens [20,78,79,80]. The present study therefore targeted the *bla*_OXA-51-like_ gene of *A. baumannii*. Findings of this study showed that the specificity of the developed genosensor was in consonance with that obtained by a conventional PCR assay. The analytical specificity of the developed genosensor assay at genomic and bacterial cell levels was tested using a panel of bacterial strains consisting of *A. baumannii* and other pathogens that commonly cause infections in humans. The capture probe immobilized on the SPCE surface was proven to be 100% specific for the *A. baumannii* target gene (*bla*_OXA-51-like_ gene), where the hybridisation process was successful and an amperometric current signal was produced. The use of the *zot* gene as an internal amplification control both for the PCR amplifications and amperometric readings in this study, eliminates the possibility of false negatives that could arise due to the presence of inhibitors in the sample. Other previously reported studies [41,42,43,45] that developed different biosensors for *A. baumannii* detection did not include any internal controls. 

The diagnostic evaluation of the developed enzyme-based genosensor assay was performed using blood samples spiked with clinical isolates. Bloodstream infection in patients due to *A. baumannii* has been commonly reported [81,82] and because of this, blood samples are widely used in PCR assay. The most common problem in the detection of pathogens in a blood sample using PCR assay is that it includes the presence of inhibitory substances such as natural components of blood [83]. As such, in this study, a blood sample inoculated in a blood culture medium was used to minimise the inhibitory substances present in the clinical samples as well as to enhance bacterial growth. The diagnostic performance for the developed enzyme-based electrochemical sequence-specific biosensor assay using aPCR amplicons was shown to be 100% sensitive and specific, with 100% positive predictive value (PPV), and negative predicted value (NPV). These findings potentially establish that the developed biosensor can still function efficiently when blood samples are used.

In addition, an accelerated stability evaluation of this developed enzyme-based electrochemical sequence-specific biosensor assay was carried out to determine the shelf-life of the assay. Trehalose as a stabiliser was used to retain the activity of protein (streptavidin) and DNA (capture probe) on the SPCE surface for the IC and target gene, respectively. Trehalose interacted with DNA by forming a glassy intermediate, thereby reducing the fluctuations of the DNA structure. In addition, during lyophilisation, trehalose could stabilize protein by making it more rigid [84]. Findings further showed that modified SPCE surfaces treated with 6% trehalose showed more relatively stable amperometric signals at all tested temperatures over the 28-day period, than those treated with 3% trehalose. In this study, the SPCE stored at 25 °C and 37 °C maintained its performance for up to 28 days. The calculated accelerated stability of the assay was determined to be 1.6 years, when stored at 4 °C. A similar stability duration was also reported by Dash and colleagues [85] where the genosensor was stable for approximately 1 year at 4 °C. The result from this study further suggests that the modified SPCE could be stored and transported without a cold chain requirement. Hence, it would be efficient for use in tropical environments as well. Other studies that reported similar electrochemical biosensors for *A. baumannii* detection [41,42,43,45] did not perform any stability evaluations to determine the performance of the method under varied temperatures.

The electrochemical genosensor developed previously and in this study could serve as an easier alternative to the laborious process of agarose gel electrophoresis and conventional culture methods in detecting *A. baumannii* in clinical specimens. However, improvements in their performance and properties to provide more powerful, miniaturized, user-friendly, and highly sensitive biosensors for nucleic acid detection would greatly impact public health. For instance, an increase in capture efficiency will also translate to a higher sensitivity [86]. Although biosensors hold great potentials for more rapid disease diagnosis, a few limitations have been reported. For instance, most reported biosensors are not multiplexable [87]. However, the integration of nanomaterials into biosensors development and integration with other technologies hold much promise towards solving this challenge [88,89]. Generally, the use of antibodies in biosensors can face limitations such as loss of biological activities due to the immobilization step. Additionally, at high density, steric hinderances can result in a loss of activity [90]. Despite this limitation, many studies have used antibodies as biorecognition molecules in electrochemical biosensor development because the method allows for easier sample preparation and also possesses good sensitivity and specificity [37]. The use of other bioreceptors such as enzymes, phages, cells, and aptamers have also been explored. 

## 5. Conclusions

This study successfully developed an enzyme-based electrochemical genosensor assay for the detection of *A. baumannii* with an internal control from the *zot* gene, which could be useful for early and specific detection of infections. Using amplicons generated from aPCR amplification of the *bla*_OXA-51-like_ gene of *A. baumannii*, the electrochemical genosensor with SPCE accurately detected the presence of complementary sequence *A. baumannii* DNA, using the hybridization principle with the help of a sequence-specific probe immobilized on the surface of the working electrode. The genosensor showed a 10-fold higher sensitivity than conventional PCR and 100% specificity with no false results or cross-reactivity among non-*A. baumannii* strains tested. Although previous works reported a lower detection limit, the LoD reported in this study (0.5 pg) is within an acceptable level for *A. baumannii* detection, and is, as well, higher than other reported LoDs. Interestingly, the developed electrochemical genosensor retained a stable performance when stored at higher temperatures of 25 °C and 37 °C, making it suitable for use in both temperate and nontemperate climates. The use of screen-printed electrodes in this study presents the advantage of adaptability, ease of mass production, and selective specificity for the target gene, making it suitable for rapid analysis on-site. The electrochemical genosensor developed in this study combines sensitivity, specificity, ease of use, and rapidity with the potential for POCT adaptability and miniaturization; hence, it can eventually be helpful to clinicians in providing prompt and appropriate treatment and management of *A. baumannii* infections in patients. With the rapid detection of *A. baumannii* in clinical specimens, the spread of *A. baumannii* can be significantly prevented, thereby improving patient health and outcomes.

## Figures and Tables

**Figure 1 microorganisms-10-01413-f001:**
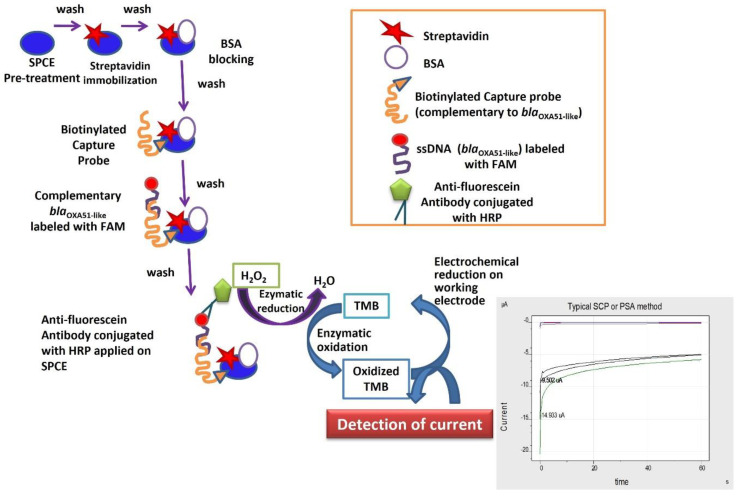
A schematic diagram of the developed electrochemical genosensor assay for detection of *A. baumannii*.

**Figure 2 microorganisms-10-01413-f002:**
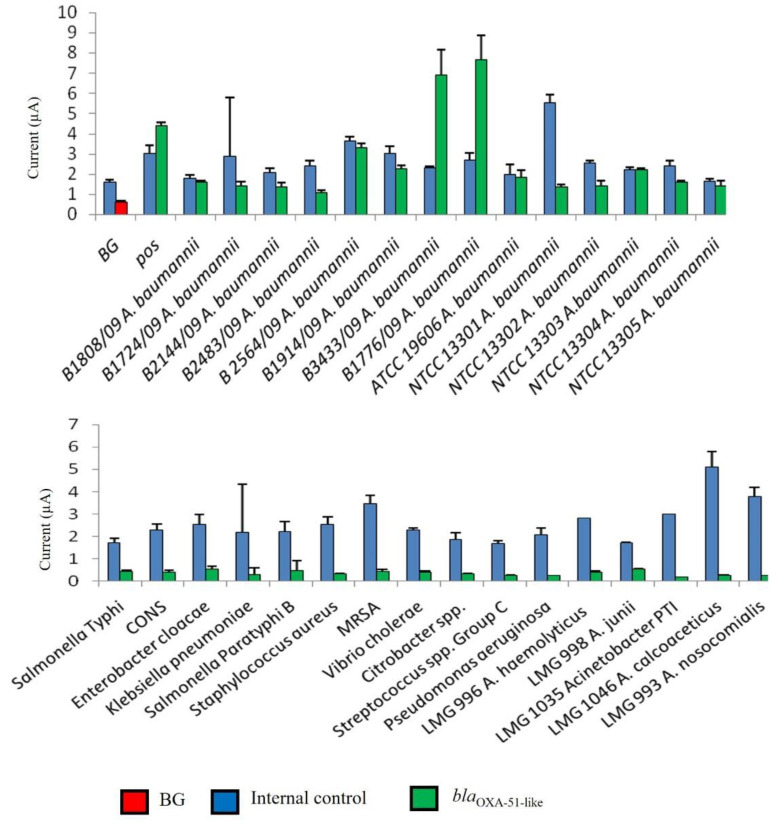
Analytical specificity of an enzyme-based electrochemical DNA biosensor using different bacterial strains. The error bars show the standard deviation for triplicate tests. CoNS—Coagulase-negative *Staphylococcus*; MRSA—Methicillin-resistant *S. aureus*; BG—background control without using bacterial strains.

**Figure 3 microorganisms-10-01413-f003:**
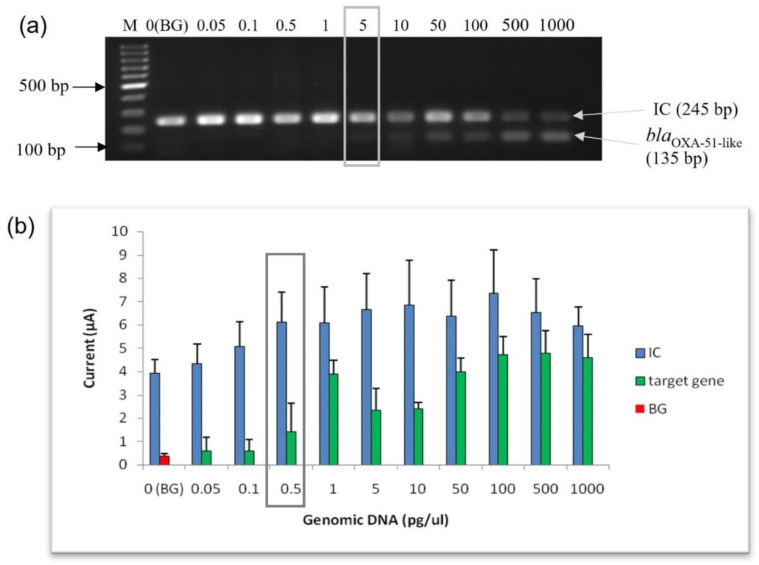
Evaluation of the analytical sensitivity of the enzyme-based electrochemical DNA biosensor using purified genomic DNA from *A. baumannii* ATCC 19606. (**a**) Agarose gel analysis; (**b**) enzyme-based DNA assay electrochemical DNA biosensor. The error bars show the standard deviation of triplicate tests. The limit of detection for the agarose gel analysis and the enzyme-based DNA assay are highlighted in the gray box. (M: 100 bp DNA ladder; BG: background control).

**Figure 4 microorganisms-10-01413-f004:**
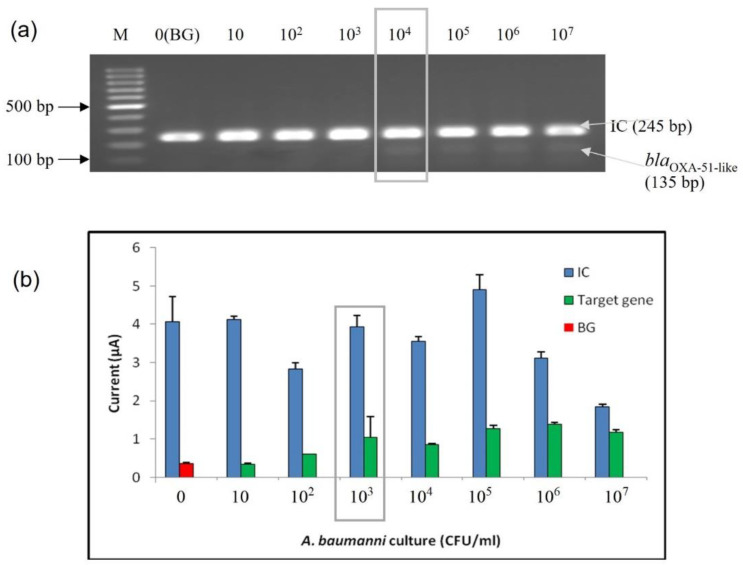
Analytical sensitivity evaluation of the enzyme-based electrochemical DNA biosensor using bacterial lysate from *A. baumannii* (ATCC 19606). (**a**) Agarose gel analysis; (**b**) enzyme-based DNA assay electrochemical DNA biosensor. The error bars show the standard deviation for triplicate tests. The limit of detection for the agarose gel and enzyme-based DNA assay were highlighted in the gray box. (M: 100 bp DNA ladder; BG: background control).

**Figure 5 microorganisms-10-01413-f005:**
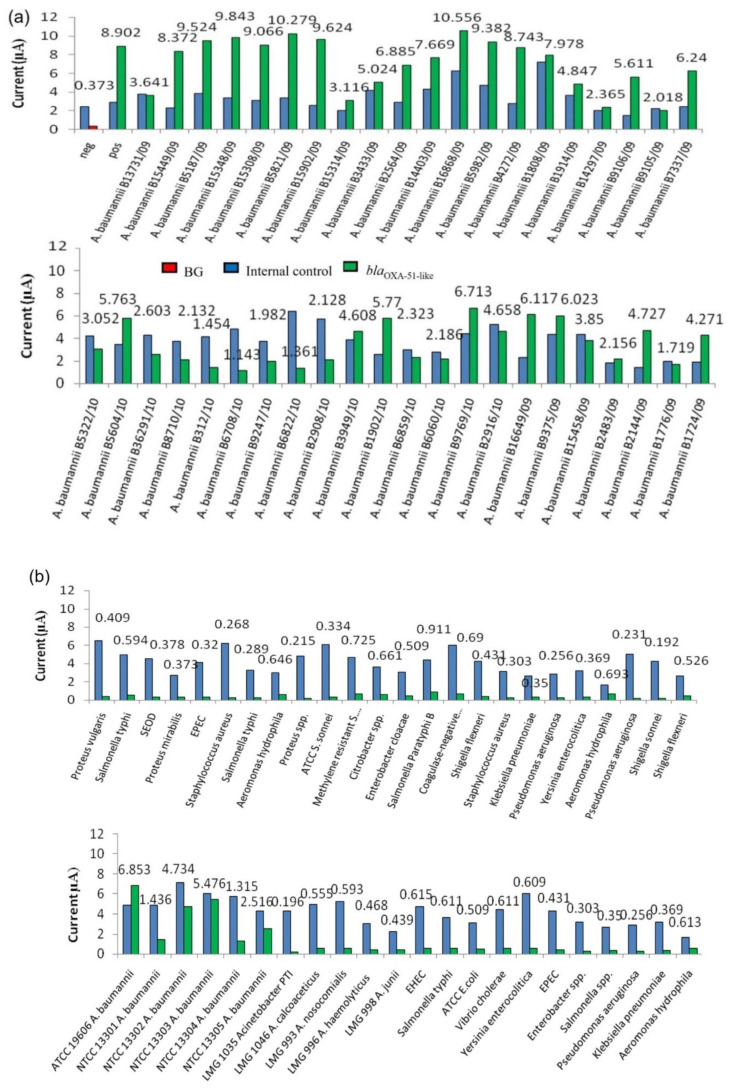
(**a**) Diagnostic evaluation of the enzyme-based electrochemical DNA biosensor using different *A. baumannii* clinical strains. (**b**) Diagnostic evaluation of the enzyme-based electrochemical DNA biosensor using different *A. baumannii* reference strains and other non-*Acinetobacter* pathogens.

**Figure 6 microorganisms-10-01413-f006:**
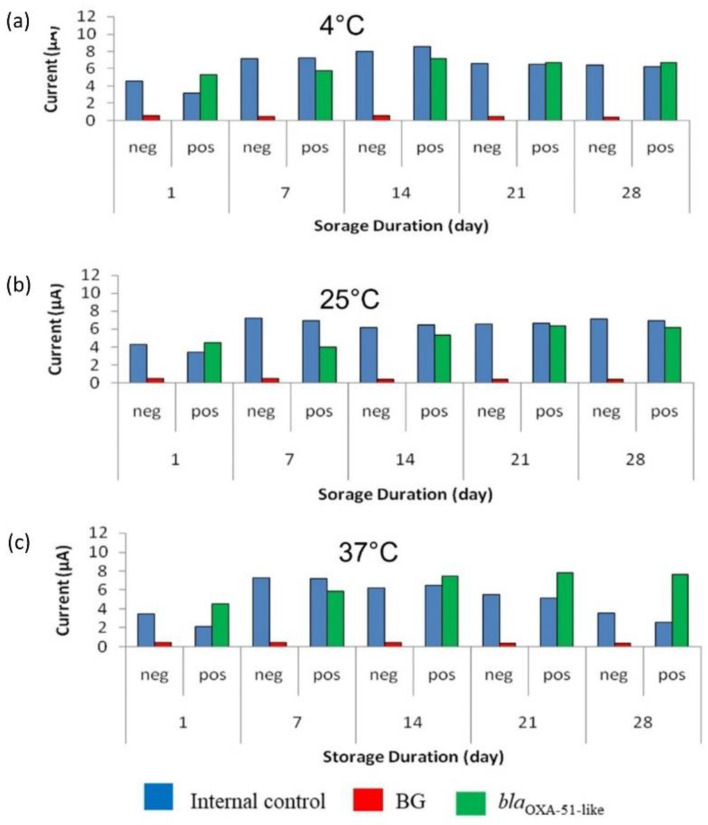
Stability evaluation of the modified screen-printed carbon electrodes (SPCEs) with 6% trehalose for 28 days at (**a**) 4 °C, (**b**) 25 °C, and (**c**) 37 °C. BG: Background control without using bacterial strains.

**Table 1 microorganisms-10-01413-t001:** List of reference strains used in this study.

Reference Strain	Quantity	Source
*Acinetobacter baumannii* ATCC 19606	1	American Type Culture Collection (ATCC)
*Shigella sonnei* ATCC 25931	1	
*Acinetobacter baumannii* NCTC13301	1	National Collection Type Cultures (NCTC)
*Acinetobacter baumannii* NCTC13302	1	
*Acinetobacter baumannii* NCTC13303	1	
*Acinetobacter baumannii* NCTC13304	1	
*Acinetobacter baumannii* NCTC13305	1	
*Acinetobacter calcoaceticus* LMG 1046	1	Belgian Coordinated Collections of Microorganisms (BCCM™/LMG)
*Acinetobacter nosocomialis* LMG 993	1	
*Acinetobacter haemolyticus* LMG 996	1	
*Acinetobacter junii* LMG 998	1	
*Acinetobacter* genospecies 3 LMG 1035	1	
Total	12	

**Table 2 microorganisms-10-01413-t002:** List of clinical strains used in this study.

Clinical Strain		Source
	Quantity	Department of Medical Microbiology and Parasitology, Universiti Sains Malaysia
*Acinetobacter baumannii*	42	
Other Bacteria:	Quantity	
Gram-Positive Bacteria		
Methicillin-Resistant *Staphylococcus aureus* (MRSA)	1	
*Staphylococcus aureus*	2	
*Streptococcus* spp.	1	
Gram-Negative Bacteria	Quantity	
*Aeromonas hydrophila*	3	
*Citrobacter* spp.	1	
Coagulase-negative *Staphylococcus* (CoNS)	1	
*Escherichia coli*	1	
Enteropathogenic *E. coli* (EPEC)	2	
Enterohaemorrhagic *E. coli* (EHEC)	1	
*Enterobacter cloacae*	1	
*Enterobacter* spp.	1	
*Klebsiella pneumoniae*	2	
*Pseudomonas aeruginosa*	2	
*Proteus mirabilis*	1	
*Proteus vulgaris*	1	
*Proteus* spp.	1	
*Salmonella* spp.	1	
*Salmonella* Paratyphi B	1	
*Salmonella* Typhi	3	
*Serratia* spp.	1	
*Shigella flexneri*	2	
*Shigella sonnei*	1	
*Vibrio cholerae*	1	
*Yersinia enterocolitica*	2	
Total	76	

**Table 3 microorganisms-10-01413-t003:** Details of oligomers used in this study.

Oligomers	Sequence	Gene	Amplicon (bp)
*bla*_OXA-51__F (forward primer)	5′-TTT AGC TCG TCG TAT TGG ACT TGA-3′	*bla* _OXA-51-like_	135
*bla*_OXA-51__R (reverse primer)	5′-/56-FAM/GCC TCT TGC TGA GGA GTA ATT TTT-3′		
Capture probe	5′-/5Bio/TGG CAA TGC AGA TAT CGG TAC CCA AGT C-3′		
Synthetic *bla*_OXA-51-like_	5′/56FAM/ GCCTCTTGCTGAGGAGTAATTTTTAAAGAATTATCGACTTGGGTACCGATATCTGCATTGCCATAACGAGTTCAAGTCCAATACGACGAGCTAAA		
IC forward primer	5′-/5Bio/AGG CGG TTG CTC CTG CGT CTT TT -3′	*zot* (IC)	245
IC reverse primer	5′-/56-FAM/CGG TAA CGG TAG CAC CTT GTA G -3′		

IC: Internal control.

## Data Availability

The data presented in this study are available on request from the corresponding author.

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
