# Peer review of "Sequence-Specific Electrochemical Genosensor for Rapid Detection of blaOXA-51-like Gene in Acinetobacter baumannii"

_microorganisms, 2022, doi:10.3390/microorganisms10071413_

Round 1
Reviewer 1 Report
The manuscript presents an interesting approach to the detection of A. baumannii; the quantity and quality of experiments are adequate for publication; however, some minor points should be addressed before acceptance: 1) the introduction should include previous related work, either for mo.o detection or in the development of similar methods; and should clearly indicate the knowledge gap to be addressed, limitations or advantages of this type of method.
2) The methodology should explain in better detail the principle of detection; lines 104-123 are not clear enough; in fact, lines 351-364 are more appropriate, and should be placed in the methods section.
3) Some figures do not have error bars. In my opinion, Table 4 is superfluous, and the result can be described with text. The same for figures 6 and 7; I think the important result of them is only the data at 28 days.
4) The discussion is minimal in content although the description is long; the authors should focus the discussion on contrasting their result with previous ones, not just describing them; the limitations of these methods in general and in particular should be discussed.
5) The conclusion is written more as an overview of what the method can achieve; I recommend focusing more on the results without being a summary of them.
Author Response
Dear Reviewer,
We are sincerely grateful for the time and effort you committed to making our manuscript better. No doubt, the very thoughtful observations made and the invaluable comments and advise you offered has given our manuscript a better structure and richer content.

Reviewer 2 Report
1. Line 26: 103 CFU/ml should be corrected to 10^3 CFU/ml
2. Line 28: the LOD was defined as 0.008 µM in the Abstract. However, it is not defined in the main text of the manuscript.
3. Line 30: the degree C symbol should be corrected
4. Line 47 onward: the common Latin words, e.g., et al., should no longer be italicized.
5. The authors should list the primer sequences for the aPCR and explain how this method generates single-stranded DNA target sequences instead of double-stranded PCR amplicons.
6. The utilization of the zot gene as an internal control is unclear. Were individual test organisms transformed with IC plasmid?
7. If purified IC plasmid DNAs were used as internal controls, does that mean IC plasmids and target DNA samples were mixed in the same test vials while taking the measurements? If not, how was it defined as an internal control?
8. How were the ICs standardized? If the ICs were standardized, why were the current (µA) readings varied greatly among test samples?
9. Figure 2 is probably best presented as a table with the amperometric currents listed for individual samples along with the p values when compared to the cut-off value.
10. For Figure 3, it is unclear why the concentration of amplicons for IC decreased with increased genomic DNA levels.
11. The authors stated that the LoD for convention PCR to be 5 pg based on Figure 3a. How was that determined? Was it based on a visual determination of the gel? Why not use qPCR for a more precise comparison?
12. The authors stated the LoD for the enzyme-based electrochemical assay to be 0.5 pg with amperometric current of 1.443 µA. However, the authors should list the associated p value as it is not apparent that the reading is statically different from that of the BG.
13. Similarly, the same comments, i.e., 11, and 12, apply to Figure 4.
14. Figure 5 should probably be presented as a table as well.
15. How does the assay perform in a mixed culture condition?
16. The authors should summarize the time required to perform the assay per sample taking into consideration the feasibility of high throughput adaptation.
a. Is simultaneous electrochemical current measurement feasible? What would be the time required to process 96 or 384 samples?
b. What is the specific instrument required to take the readings? The multiple steps required for the preparation of individual screen-printed carbon electrodes could be a bottleneck for the technology, especially for high throughput applications.
c. The use of antibodies in the assays could be a disadvantage of the technology.
d. For conventional PCR, 384 samples per run is possible; colony PCR without the need for DNA extraction is also feasible.
e. The authors should discuss and compare these advantages and disadvantages.
17. What is the required sample volume?
Author Response
Dear Reviewer,
We are grateful for the time and effort you put into making our manuscript better. No doubt, the very thoughtful observations made and the invaluable comments and advise have contributed to giving our manuscript a better structure and richer content.
